# Decoupling Gating from Linearity

## Abstract

The gap between the empirical success of deep learning and the lack of strong theoretical guarantees calls for studying simpler models. By observing that a ReLU neuron is a product of a linear function with a gate (the latter determines whether the neuron is active or not), where both share a jointly trained weight vector, we propose to decouple the two. We introduce GaLU networks — networks in which each neuron is a product of a Linear Unit, defined by a weight vector which is being trained, with a Gate, defined by a different weight vector which is not being trained. Generally speaking, given a base model and a simpler version of it, the two parameters that determine the quality of the simpler version are whether its practical performance is close enough to the base model and whether it is easier to analyze it theoretically. We show that GaLU networks perform similarly to ReLU networks on standard datasets and we initiate a study of their theoretical properties, demonstrating that they are indeed easier to analyze. We believe that further research of GaLU networks may be fruitful for the development of a theory of deep learning.

## 1 Introduction

An artificial neuron with the ReLU activation function is the function $f_{\boldsymbol{w}}(\boldsymbol{x}) : \mathbb{R}^d \to \mathbb{R}$ such that

$$f_{\boldsymbol{w}}(\boldsymbol{x}) = \max\{\boldsymbol{x}^\top \boldsymbol{w}, 0\} = \left(\mathbf{1}_{\boldsymbol{x}^\top \boldsymbol{w} \geq 0}\right) \cdot \left(\boldsymbol{x}^\top \boldsymbol{w}\right) \ .$$

The latter formulation demonstrates that the parameter vector $\boldsymbol{w}$ has a dual role; it acts both as a *filter* or a *gate* that decides if the neuron is active or not, and as *linear weights* that control the value of the neuron if it is active. We introduce an alternative neuron, called *Gated Linear Unit* or GaLU for short, which decouples between those roles. A $0-1$ GaLU neuron is a function $g_{\boldsymbol{w},\boldsymbol{u}}(\boldsymbol{x}) : \mathbb{R}^d \to \mathbb{R}$ such that

$$g_{\boldsymbol{w},\boldsymbol{u}}(\boldsymbol{x}) = \left(\mathbf{1}_{\boldsymbol{x}^\top \boldsymbol{u} \geq 0}\right) \cdot \left(\boldsymbol{x}^\top \boldsymbol{w}\right) \ . \tag{1}$$

GaLU neurons, and therefore GaLU networks, are at least as expressive as their ReLU counterparts, since $f_{\boldsymbol{w}} = g_{\boldsymbol{w},\boldsymbol{w}}$. On the other hand, GaLU networks appear problematic from an optimization perspective, because the parameter $\boldsymbol{u}$ cannot be trained using gradient based optimization (since $\nabla_{\boldsymbol{u}} g_{\boldsymbol{w},\boldsymbol{u}}(\boldsymbol{x})$ is always zero). In other words, training GaLU networks with gradient based algorithms is equivalent to initializing the vector $\boldsymbol{u}$ and keeping it constant thereafter. A more general definition of a GaLU network is given in section 2.

The main claim of the paper is that GaLU networks are on one hand as effective as ReLU networks on real world datasets (section 3) while on the other hand they are easier to analyze and understand (section 4).

### 1.1 Related Work

Many recent works attempt to understand deep learning by considering simpler models, that would allow theoretical analysis while preserving some of the properties of networks of practical utility. Our model is most closely related to two such proposals: linear networks and non-linear networks in which only the readout layer is being trained.

Deep linear networks is a popular model for analysis that lead to impressive theoretical results (e.g. Saxe et al. (2013); Kawaguchi (2016); Lu & Kawaguchi (2017)). Linear networks are useful in order to understand how well gradient-based optimization algorithms work on non-convex problems. The

weakness of linear network is that their expressive power is very limited: linear networks can only express linear functions. It means that their usefulness to understand the practical success of standard networks is somewhat limited.

Training only the readout layer is an alternative attempt to understand deep learning through simpler models, that also gave theoretical interesting results (e.g. Saxe et al. (2011); Mairal et al. (2014); Daniely et al. (2016)). The idea is that all the layers but the last one implement a non-linear constant transformation, and the last layer is learning a linear function on top of this transformation. The weakness of this model is that there is a big practical difference between training all the layers of a network and training only the last one.

Our model is similar in certain aspects to both of those models, but it enjoys a much better practical utility than either one. See section 3 for an empirical comparison.

## 2 GaLU Networks

Recall the definition of a basic GaLU neuron given in equation 1. We consider a more general GaLU neuron of the form

$$g_{\boldsymbol{w},\boldsymbol{u},\sigma}(\boldsymbol{x}) = \sigma(\boldsymbol{x}^\top \boldsymbol{u}) \cdot \left(\boldsymbol{x}^\top \boldsymbol{w}\right)$$

for some non-linear scalar function $\sigma : \mathbb{R} \to \mathbb{R}$. If $\sigma$ is differentiable, we could train the vectors $\boldsymbol{u}$ with gradient based algorithms, but the focus of this paper is on untrained gates. That is, we assume that the vectors $\{\boldsymbol{u}\}$ are kept to their initial values throughout the optimization procedure and only the linear part of the GaLU neurons is being optimized.

GaLU networks with a single hidden layer have the following property: for any given example, the values of the gates in the network remain constant. In networks with more than one hidden layer this not true. Consider a standard fully connected feed-forward network, let $\boldsymbol{x}^{(0)}$ be the input to the network and let $\boldsymbol{x}^{(1)}, \boldsymbol{x}^{(2)}, \ldots$ be the inputs to intermediate layers of the network. The output of a GaLU neuron at layer $i$ will be $\sigma(\boldsymbol{x}^{(i-1)\top} \boldsymbol{u}) \cdot \left(\boldsymbol{x}^{(i-1)\top} \boldsymbol{w}\right)$. So while the filter parameter vector, $\boldsymbol{u}$, is not optimized upon, the value of the gate, $\sigma(\boldsymbol{x}^{(i-1)\top} \boldsymbol{u})$, can change as $\boldsymbol{x}^{(i-1)}$ changes. This adds an additional complication to the dynamics of the optimization that we wish to avoid.

An alternative way to define a GaLU neuron at layer $i$ is $\sigma(\boldsymbol{x}^{(0)\top} \boldsymbol{u}) \cdot \left(\boldsymbol{x}^{(i-1)\top} \boldsymbol{w}\right)$. In that case, the value of the gate is determined by the original input, and only the linear part depends on the output of the previous layer of the network. We call such a neuron a *GaLU0* neuron, and a GaLU0 *network* is a network where all the neurons are GaLU0 neurons. In GaLU0 networks the gate values remain constant along the training, producing simpler dynamics.

## 3 Empirical Success

In order to check the hypothesis that effectiveness of ReLU networks stems mostly from the ability to train the linear part of the neurons, and not the gate part, we tested both GaLU0[1] and ReLU networks on the standard MNIST (LeCun & Cortes, 2010) and Fashion-MNIST (Xiao et al., 2017) datasets. For both, we used PCA to reduce the input dimension to 64, and then trained a two hidden layers fully-conneted networks on them, with $k$ hidden neurons at each hidden layer. Figure 1 summarizes the results, showing that GaLU0 and ReLU achieve similar results, both outperforming linear networks of the same size. Training only the readout layer of a ReLU network gave much poorer results (which were omitted from the graphs for clarity).

## 4 Theoretical Simplicity: Solving $\mathbb{R}^d \to \mathbb{R}^1$ Problems with One Hidden Layer Networks

We now turn to some very basic theoretical analysis of GaLU networks with a single hidden layer. Our goal is to show that GaLU networks are simpler to analyze than standard networks.

---

[1]We used the logistic sigmoid function, as it gave better results on the test set than the binary gate function.

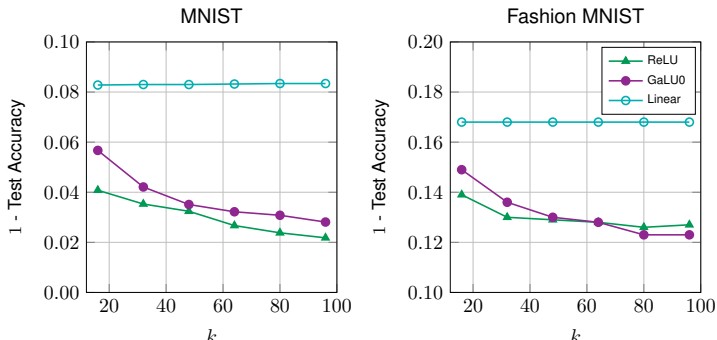

Figure 1: Comparison between 3 deep learning models on the MNIST and Fashion- MNIST datasets. All models were trained using the same architectures: two fully connected hidden layers with $k$ neurons. The input dimension was reduced to 64 with PCA.

Consider a GaLU network with a single hidden layer of $k$ neurons: $\mathcal{N}(\boldsymbol{x}) = \sum_{j=1}^{k} \alpha_j g_{\boldsymbol{w}_j, \boldsymbol{u}_j}(\boldsymbol{x})$. A convenient property of a GaLU neuron is that it is linear in the weights $w_j$, hence, $\alpha_j g_{\boldsymbol{w}_j, \boldsymbol{u}_j}(\boldsymbol{x}) = g_{\alpha_j \boldsymbol{w}_j, \boldsymbol{u}_j}(\boldsymbol{x})$. It means that the network can be rewritten as

$$\mathcal{N}(\boldsymbol{x}) = \sum_{j=1}^{k} \alpha_j g_{\boldsymbol{w}_j, \boldsymbol{u}_j}(\boldsymbol{x}) = \sum_{j=1}^{k} g_{\alpha_j \boldsymbol{w}_j, \boldsymbol{u}_j}(\boldsymbol{x}) = \sum_{j=1}^{k} g_{\tilde{\boldsymbol{w}}_j, \boldsymbol{u}_j}(\boldsymbol{x})$$

with $\tilde{\boldsymbol{w}}_j = \alpha_j \boldsymbol{w}_j$. Because we want to optimize over the weights $\boldsymbol{w}_1, \ldots, \boldsymbol{w}_k, \alpha_1, \ldots, \alpha_k$, we might as well optimize over the reparameterization $\tilde{\boldsymbol{w}}_1, \ldots, \tilde{\boldsymbol{w}}_k$ without losing expressive power. It means that in a GaLU network of this form, it is sufficient to train the *first* layer of the network, as the readout layer adds nothing to the expressiveness of the network.

The previous term can be further simplified:

$$\mathcal{N}(\boldsymbol{x}) = \sum_{j=1}^{k} g_{\boldsymbol{w}_j, \boldsymbol{u}_j}(\boldsymbol{x}) = \sum_{j=1}^{k} \sigma\left(\boldsymbol{x}^\top \boldsymbol{u}_j\right) \boldsymbol{x}^\top \boldsymbol{w}_j$$

$$= \begin{bmatrix} \sigma\left(\boldsymbol{x}^\top \boldsymbol{u}_1\right) \boldsymbol{x}^\top & \sigma\left(\boldsymbol{x}^\top \boldsymbol{u}_2\right) \boldsymbol{x}^\top & \ldots & \sigma\left(\boldsymbol{x}^\top \boldsymbol{u}_k\right) \boldsymbol{x}^\top \end{bmatrix} \begin{bmatrix} \boldsymbol{w}_1 \\ \boldsymbol{w}_2 \\ \vdots \\ \boldsymbol{w}_k \end{bmatrix}$$

$$= \Phi_{\boldsymbol{u}}(\boldsymbol{x})^\top \boldsymbol{w}$$

where

$$\Phi_{\boldsymbol{u}}(\boldsymbol{x}) = \begin{bmatrix} \sigma\left(\boldsymbol{x}^\top \boldsymbol{u}_1\right) \boldsymbol{x} \\ \sigma\left(\boldsymbol{x}^\top \boldsymbol{u}_2\right) \boldsymbol{x} \\ \vdots \\ \sigma\left(\boldsymbol{x}^\top \boldsymbol{u}_k\right) \boldsymbol{x} \end{bmatrix}, \quad \boldsymbol{w} = \begin{bmatrix} \boldsymbol{w}_1 \\ \boldsymbol{w}_2 \\ \vdots \\ \boldsymbol{w}_k \end{bmatrix}, \quad \boldsymbol{u} = \begin{bmatrix} \boldsymbol{u}_1 & \boldsymbol{u}_2 & \ldots & \boldsymbol{u}_k \end{bmatrix}.$$

So it turns out that a GaLU network is nothing more than a random non-linear transformation $\Phi_{\boldsymbol{u}} : \mathbb{R}^d \to \mathbb{R}^{kd}$ and then a linear function.

### 4.1 EXPRESSIVITY

There are different notions for the expressivity of a model, and one of the simplest ones is the finite-sample expressivity over a random sample. This notion fits well to our model, because we are not interested in the absolute expressivity of a GaLU network, but of the expressivity of a GaLU network with random filters. So the question is how well does a randomly-initialized network can fit a random sample. Note that given the constant filters, solving for the best weights is a convex problem.

Hence, there is no "expressivity – optimization gap" in GaLU networks – every expressivity results is immediately also an optimization result.

Let $S = \{(\mathbf{x}_1, \mathbf{y}_1), (\mathbf{x}_2, \mathbf{y}_1), \ldots, (\mathbf{x}_m, \mathbf{y}_m)\}$ be a random sample, such that $\mathbf{x}_1, \ldots, \mathbf{x}_m \sim N(0, \boldsymbol{I}_d)$ and $\mathbf{y}_1, \ldots, \mathbf{y}_m \sim N(0, 1)$, all of which are independent. Clearly, it is impossible to generalize from the sample to unseen examples; the best possible test loss is $1$, and is achieved by the constant prediction $0$. However, it is an interesting problem because it allows us to measure the expressivity of GaLU networks, by showing how much overfit we can expect from the network for a non-adversarial sample. Equivalently, it tells us how well the network can perform memorization tasks, where the only solution is to memorize the entire sample. We train the network for the standard mean-square-error regression loss.

Because the network is simply linear function over a constant non-linear transformation, and because we use the MSE loss, there is a closed form solution to the optimization problem $\min_{\boldsymbol{w}} \frac{1}{m} \sum_{i=1}^{m} (\mathcal{N}(\boldsymbol{x}_i) - y_i)^2$ which is

$$\bar{\boldsymbol{X}} = \begin{bmatrix} \Phi_{\boldsymbol{u}}(\boldsymbol{x}_1)^\top \\ \Phi_{\boldsymbol{u}}(\boldsymbol{x}_2)^\top \\ \vdots \\ \Phi_{\boldsymbol{u}}(\boldsymbol{x}_m)^\top \end{bmatrix} \quad \boldsymbol{w}^* = \bar{\boldsymbol{X}}^+ \begin{bmatrix} y_1 \\ y_2 \\ \vdots \\ y_m \end{bmatrix}$$

with $\bar{\boldsymbol{X}}^+$ being a pseudo-inverse of $\bar{\boldsymbol{X}}$. This gives us

**Theorem 1** *Let $\boldsymbol{x}_1, \ldots, \boldsymbol{x}_m \in \mathbb{R}^d$, $\boldsymbol{u}_1, \ldots, \boldsymbol{u}_k \in \mathbb{R}^d$ be arbitrary vectors. Define $\bar{\boldsymbol{X}}$ as above. Let $\mathbf{y}_1, \ldots, \mathbf{y}_m \sim N(0, 1)$ be independent random normal variables. Define the expected squared loss on the training set, for weights $\boldsymbol{w}$, as $L_S(\boldsymbol{w})$. Then,*

$$\mathbb{E}[\min_{\boldsymbol{w}} L_S(\boldsymbol{w})] = 1 - \frac{rank\left(\bar{\boldsymbol{X}}\right)}{m} \ .$$

**Proof** Every vector $\boldsymbol{y} = (\mathbf{y}_1, \ldots, \mathbf{y}_m) \in \mathbb{R}^m$ can be decomposed to a sum $\boldsymbol{y} = \mathbf{a} + \mathbf{b}$ where $\mathbf{a}$ is in the span of the columns of $\bar{\boldsymbol{X}}$ and $\mathbf{b}$ is in the null space of $\bar{\boldsymbol{X}}$. It follows that $\min_{\boldsymbol{w}} L_S(w) = \|\mathbf{b}\|^2/m$. The claim follows because if $\boldsymbol{y} \sim N(0, I_m)$ then the expected value of $\|\mathbf{b}\|^2$ is $m - \text{rank}\left(\bar{\boldsymbol{X}}\right)$. ∎

It is always true that $\text{rank}(\bar{\boldsymbol{X}}) \leq \min\{m, kd\}$. Empirical experimentation shows that if $\mathbf{x}_1, \ldots, \mathbf{x}_m, \mathbf{u}_1, \ldots, \mathbf{u}_k \sim N(0, \boldsymbol{I}_d)$ then with high probability $\text{rank}(\bar{\boldsymbol{X}}) = \min\{m, kd\}$.

### 4.1.1 COMPARISON TO RELU NETWORKS

The fact that the GaLU network turned out to be only a linear function on top of a non-linear transformation seems to be a peculiar mathematical accident, with little relevance to standard networks. So we empirically tested the behavior of both ReLU and GaLU networks on the above model. It turns out that ReLU outperforms GaLU by a small margin – it is never better than GaLU with double the number of neurons, and is often worse than that.

ReLU can outperform GaLU, even though it is less expressive, because we don't train the value of the the filters $\boldsymbol{u}_1, \ldots, \boldsymbol{u}_k$ at all for the GaLU networks. It turns out that SGD over a ReLU network converges to better filters than a simple random initialization. One way to measure how much better those filters are is by trying to improve the initial filters of the GaLU network by randomly replacing them. Consider for example the simple algorithm given in algorithm 1.

Running this algorithm improves the results of the GaLU networks, making them more competitive with the ReLU ones. Figure 2 summarizes our results.

## 4.2 GENERALIZATION

An important fact about artificial neural networks is that they have small generalization error in many real-life problems. Otherwise they wouldn't be very useful as a learning algorithm. Zhang

---

**Algorithm 1** Improve GaLU filters

---

**Input:** A sample $S$, number of neurons $k$, number of iterations $n$.
    Initialize $\mathbf{u}_1, \mathbf{u}_2, \ldots, \mathbf{u}_k$ randomly.
    Find an optimal solution $\mathbf{w}_1, \ldots, \mathbf{w}_k$.
    **for** $i = 1$ **to** $n$ **do**
        Pick $j \sim \text{Uniform}\{1, 2, ..., k\}$.
        Pick $\tilde{\mathbf{u}}_j$ randomly.
        Find an optimal solution for a GaLU network with filters $\mathbf{u}_1, \ldots, \mathbf{u}_{j-1}, \tilde{\mathbf{u}}_j, \mathbf{u}_{j+1}, \ldots, \mathbf{u}_k$.
        If the new solution is better than the current one, update $\mathbf{u}_j = \tilde{\mathbf{u}}_j$.
    **end for**

---

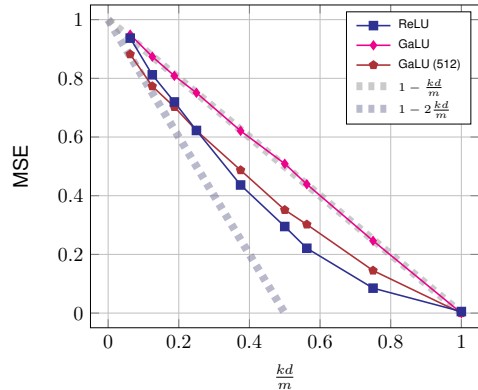

Figure 2: Comparison of GaLU and ReLU networks with a single hidden layer and output in $\mathbb{R}^1$. GaLU(512) stands for GaLU networks after 512 steps of algorithm 1.

et al. (2016) have shown empirically that many classical attempts, such as model capacity, explicit regularization and even the properties of the optimization algorithm cannot explain this behavior. One of the main experiments they run was to train the network over a sample with randomized labels, and to observe that the network still achieved small training loss (but large test loss, naturally). So any generalization bound that can be applied to the randomized sample is necessarily too weak to explain the generalization of the natural sample.

As our goal is to show that GaLU networks exhibit similar phenomena as ReLU networks, but may be easier to analyze, we first construct a similar experiment to that of Zhang et al. (2016) and compare the performance of GaLU and ReLU networks. Consider the following natural model. Let $\mathbf{c}_1, \ldots, \mathbf{c}_n \sim N(0, \boldsymbol{I}_d)$ be $n$ clusters centers, each one with a random labels $\mathbf{b}_1, \ldots, \mathbf{b}_n$. A data point $(\mathbf{x}, \mathbf{y})$ is generated by picking a random index $\mathbf{i} \sim \text{Uniform}\{1, 2, \ldots, n\}$, and setting $\mathbf{x} = \mathbf{c}_\mathbf{i} + \xi$ for $\xi \sim N(0, \sigma_x^2 I_d)$. $\mathbf{y}$ is a noisy version of $\mathbf{b}_\mathbf{i}$. This model can be used for both regression problems (with $\mathbf{b}_i \sim N(0, 1)$ and $\mathbf{y} = \mathbf{b}_\mathbf{i} + \epsilon$, $\epsilon \sim N(0, \sigma_y^2)$) and classification problems (with $\mathbf{b}_i \sim \text{Ber}(\frac{1}{2})$, and $\mathbf{y} = \mathbf{b}_\mathbf{i} \oplus \epsilon$, $\epsilon \sim \text{Ber}(p)$).

We fixed the number of samples $m = 1000$, the input dimension $d = 30$, the number of clusters $n = 30$, the number of hidden neurons $k = 30$ and $\sigma_x = 0.1$. We calculated the train and test errors for different values of $\sigma_y$ and $p$ and for a GaLU and ReLU networks. The results are summarized in figure 3. We can clearly see that GaLU and ReLU have similar statistical behavior, and that while the train error is always small, as the labels become noisier the generalization error increases. This matches the spirit of experiments reported in Zhang et al. (2016).

### 4.2.1 GENERALIZATION ERROR AND NORMS

Next, we turn to an analysis of this phenomenon. Since one hidden layer GaLU networks can be cast as linear predictors, we can rely on classic norm-based generalization bounds for linear predictors. In particular, for $p \in \{1, 2\}$, consider the class of linear predictors $\mathcal{H}_p = \{\boldsymbol{x} \mapsto \boldsymbol{x}^\top \boldsymbol{w} : \|\boldsymbol{w}\|_p \leq B_p\}$. Given a sequence of instances $S = \{\boldsymbol{x}_1, \ldots, \boldsymbol{x}_m\}$, where $\|\boldsymbol{x}_i\|_\infty \leq 1$, the Rademacher complexity

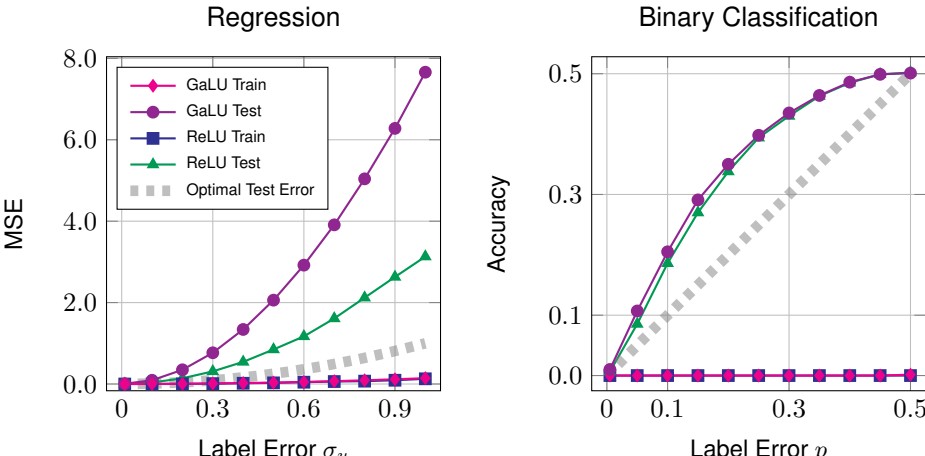

Figure 3: Train and test errors for the two models from section 4.2, with $n = k = d = 30, m = 1000, \sigma_x = 0.1$. Both of those graphs show that the generalization error is highly correlated with the optimal error: it is not true that there is a constant difference between the train error and test error. Note that in the regression problem, the number of SGD steps can drastically change the test error. More steps mean larger test error. Similar optimization issues might also account for the apparent difference between GaLU and ReLU in the regression model.

of all the predictions $\mathcal{H}_p$ induces on $S$ is upper bounded by $\sqrt{B_2^2 \max_i \|\boldsymbol{x}_i\|_2^2 / m}$ for $p = 2$ and by $\sqrt{2 \log(2d) B_1^2 \max_i \|\boldsymbol{x}_i\|_\infty^2 / m}$, where $d$ is the dimension, for $p = 1$. See for example Section 26.2 in Shalev-Shwartz & Ben-David (2014). This also induces an upper bound on the gap between the test and train loss (see again Shalev-Shwartz & Ben-David (2014) for Lipschitz loss functions and see Srebro et al. (2010) for the relation between Rademacher complexity and the generalization of smooth losses such as the squared loss). The question is whether the $\ell_1/\ell_2$ norm of $\boldsymbol{w}$ is correlated with the amount of noise in the data. To study this, we depict the gap between train and test error as a function of the norm of $\boldsymbol{w}$ for GaLU networks. As can be seen in figure 4, for both the $\ell_1$ and $\ell_2$ norm, there is a clear linear relation between $\|\boldsymbol{w}\|_p^2$ and the generalization gap. While the constants are far from what the bounds state, the linear correlation is very clear.

Note that figure 4 deals with GaLU networks that were trained as linear functions (by using the closed form solution for the MSE loss), and indeed shows that such network with such training behave as the theory states for linear predictors. We do not get the same behavior when we (unnecessarily) train both layers of the network using SGD. This matches the discussion in Section 5 of Zhang et al. (2016), where the correlation between the $\ell_2$ norm of the weights in a ReLU network and the test loss is discussed, and it is argued that there are more factors that affect the generalization properties. Indeed, many followup works show different capacity measures that may be more adequate for studying the generalization of deep learning (See for example Bartlett et al. (2017); Neyshabur et al. (2017b; 2018); Arora et al. (2018); Neyshabur et al. (2017a); Kawaguchi et al. (2017)). We next show a rather different analysis for a particular instance of linear regression.

### 4.2.2 ALTERNATIVE APPROACH

Consider a simple linear regression using the MSE, and denote the train and test loss by

$$L_S (\boldsymbol{w}) = \frac{1}{m} \sum_{(\boldsymbol{x}, y) \in S} \left( \boldsymbol{x}_i^\top \boldsymbol{w} - y \right)^2 \quad ; \quad L_\mathcal{D} (\boldsymbol{w}) = \mathbb{E}_{(\mathbf{x}, \mathbf{y}) \sim \mathcal{D}} \left( \mathbf{x}^\top \boldsymbol{w} - \mathbf{y} \right)^2 .$$

Given a training set $S$, the MSE estimator is defined as $\boldsymbol{w}(S) := \arg \min_{\boldsymbol{w}} L_S(\boldsymbol{w})$.

We start with the following lemma.

**Lemma 1 (Follows from Corollary 2 of Rosset & Tibshirani (2018))** *For a scalar $\sigma \geq 0$ and a vector $\beta \in \mathbb{R}^d$, let $\mathcal{D}_{\sigma,\beta}$ be the distribution over $\mathbb{R}^d \times \mathbb{R}$ which is defined by the following generative*

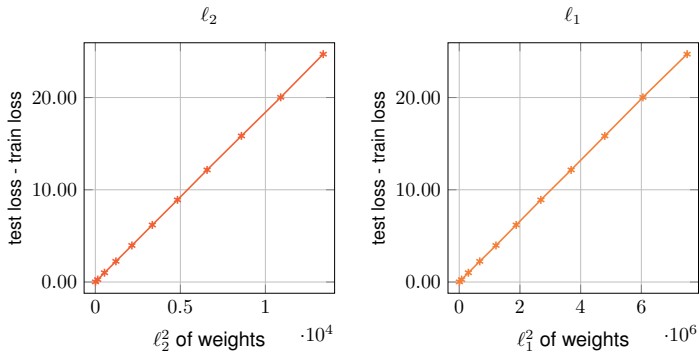

Figure 4: A linear correlation between the generalization gap and the squared norm of the solution. This was generated for GaLU network with a fixed readout layer, and the solution was calculated analytically.

*model: pick $\mathbf{x} \sim N(0, \boldsymbol{I}_d)$, and pick $\mathbf{y} = \mathbf{x}^\top \beta + \sigma \varepsilon$ where $\varepsilon \sim N(0, 1)$. Then, for $m > d + 1$, we have:*

$$\mathop{\mathbb{E}}_{S \sim \mathcal{D}_{\sigma,\beta}^m} [L_S(\mathbf{w}(S))] = \sigma^2 \left(1 - \frac{d}{m}\right)$$

$$\mathop{\mathbb{E}}_{S \sim \mathcal{D}_{\sigma,\beta}^m} \left[L_{\mathcal{D}_{\sigma,\beta}}(\mathbf{w}(S))\right] = \sigma^2 \left(1 + \frac{d}{m} + \frac{d}{m} \frac{d+1}{m-d-1}\right) .$$

This lemma provides a complete analysis for the following experiment, which is similar to the experiments reported by Zhang et al. (2016). We compare two distributions, the first is $\mathcal{D}_{\sigma,\beta}$ for some vector $\beta \in \mathbb{R}^d$ and for $\sigma$ being close to 0, and the second is $\mathcal{D}_{1,0}$. Note that the first distribution corresponds to a case in which we would like to be able to generalize, while the second distribution corresponds to a case in which we are fitting random noise and do not expect to generalize. We set the training set size to be $m = d + 2$ and we analyze the MSE estimator, $\boldsymbol{w}(S)$. As the lemma shows, the expected training losses on the first and second distributions are

$$\sigma^2 \left(1 - \frac{d}{m}\right) = \frac{2\,\sigma^2}{m} \quad ; \quad \frac{2}{m} ,$$

respectively. Hence, the training loss should be small on both of the distributions. In contrast, the expected test loss on the first distribution is

$$\sigma^2 \left(1 + \frac{d}{m} + \frac{d}{m} \frac{d+1}{m-d-1}\right) \le (3+d)\sigma^2$$

while the expected test loss on the second distribution is

$$\left(1 + \frac{d}{m} + \frac{d}{m} \frac{d+1}{m-d-1}\right) \ge 1 .$$

We see that while the train loss can be small on both distributions, in the test loss we see a big gap between the first distribution (assuming $\sigma \ll 1/\sqrt{d}$) and the second distribution of purely random labels. This is exactly the type of phenomenon reported in Zhang et al. (2016) — a sample with a small amount of noise achieves both small train and test losses, but a sample with random labels achieves a small train loss but a large test loss. Note that this is a natural property of the least squares solution, without any explicit regularization, picking a minimal-norm solution or using a specific algorithm for solving the problem.

Lemma 1 gives us a very sharp analysis of linear regression. Unfortunately, the assumptions of Lemma 1 (which are based on the assumptions of Corollary 2 in Rosset & Tibshirani (2018)) are too strong — we need that $m > d + 1$ and that the instances will be generated based on a Gaussian distribution. While Rosset & Tibshirani (2018) also includes asymptotic results that are applicable for a larger set of distributions, we leave the application of them to GaLU networks for future work.

# 5 A FEW WORDS ABOUT $\mathbb{R}^d \to \mathbb{R}^{d'}$ PROBLEMS WITH ONE HIDDEN LAYER NETWORKS

In the analysis of the $\mathbb{R}^d \to \mathbb{R}$ case we used the fact that a GaLU neuron $g_{\boldsymbol{w},\boldsymbol{u}}$ is linear in the parameter $\boldsymbol{w}$, and it allowed us to rephrase the problem as a convex problem. In the $\mathbb{R}^d \to \mathbb{R}^{d'}$ case the situation is not as simple. In this case, every hidden neuron has $d'$ outgoing edges, and so we cannot use the same reparametrization trick as before.

Even so, the output of a GaLU neuron is still linear in the parameter $\boldsymbol{w}$. It means that for convex loss functions, finding the optimal weights for the first layer, keeping the weights of the second one constant, is a convex problem. The same doesn't hold for ReLU networks. Finding the optimal weights for the second layer, keeping the weights of the first one constant, is also a convex problem. Even more specifically, the optimization problem over the two layers is biconvex (see Gorski et al. (2007) for a survey). So instead of applying SGD, we can apply biconvex optimization algorithms, such as Alternate Convex Search (ACS). In the case of the MSE loss, there is a closed form solution for each step of ACS, and using it outperforms SGD for small enough samples[2]. Even though it is of limited practical use, this algorithm might be interesting for the derivation of theoretical bounds for such networks.

In addition, it turns out that as we increase the output dimension $d'$, GaLU and ReLU networks becomes more similar. In section 4.1.1 we measured the difference between ReLU and GaLU for the problem where all the variables are i.i.d. $N(0, 1)$, and it turned out that ReLU outperforms GaLU to a small extent. We repeated this experiment with larger $d'$, and saw that the difference between the two vanished quickly (see figure 5).

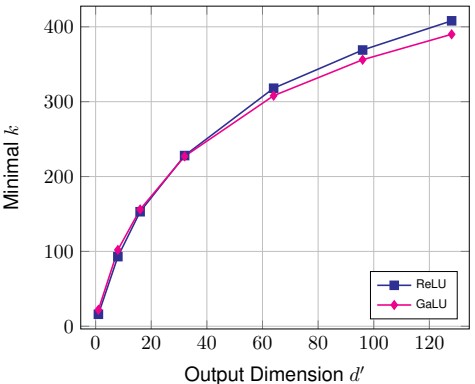

Figure 5: We empirically found the minimal number of neurons $k$ such that a one hidden layer network achieves MSE$< 0.3$ on the random regression problem. As the output dimension $d'$ grows, more neurons are needed. As demonstrated in figure 2, GaLU networks needs more neurons than ReLU networks for output dimension $d' = 1$. For larger $d'$ GaLU is slightly better, but it is clear that the two networks exhibit very similar behavior. We used fixed sample size ($m = 1024$) and input dimension ($d = 32$) in the generation of this graph.

# 6 CONCLUSION & FURTHER WORK

The standard paradigm in deep learning is to use neurons of the form $\sigma\left(\boldsymbol{x}^\top \boldsymbol{w}\right)$ for some differentiable non linear function $\sigma : \mathbb{R} \to \mathbb{R}$. In this article we proposed a different kind of neurons, $\sigma_{i,j} \cdot \boldsymbol{x}^\top \boldsymbol{w}$, where $\sigma_{i,j}$ is some function of the example and the neuron index that remains constant along the training. Those networks achieve similar results to those of their standard counterparts, and they are easier to analyze and understand.

To the extent that our arguments are convincing, it gives new directions for further research. Better understanding of the one hidden layer case (from section 5) seems feasible. And as GaLU and ReLU networks behave identically for this problem, it gives us reasons to hope that understanding the behavior of GaLU networks would also explain ReLU networks and maybe other non-linearities as well. As for deeper network, it is also not beyond hope that GaLU0 networks would allow some better theoretical analysis than what we have so far.

---

[2]As it requires inverting a matrix, it is infeasible for large samples.

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
