# OpenReview forum: "Decoupling Gating from Linearity"
_ICLR.cc/2019/Conference_

### Official Review · AnonReviewer2 · 2018-10-31
**Intellectual exercise of limited value to the community**

**Rating:** 3
**Confidence:** 5

**Review:**

The paper proposes an alternative to commonly used ReLU activated networks. The "gating" and "amount" effects of the weights are decoupled. The authors claim that such architectures are easier to theoretically understand. That might be the case indeed, but I fail to see much value in obtaining such understanding of very contrived objects that are not being used in practice. Unless such architectures can be proven to be interesting from a practical standpoint I do not think there is much of a point in studying them. The argument provided by the authors that they can - in a simple situation - have as much expressive power as a standard ReLU activated architecture is insucfficient, in my opinion, to justify researching them. Also, if a strong, deep theorem was proven using GaLU networks was proven, I would be inclined to recommend the paper to be accepted. As is - I do not find the paper to be a contribution significant enough for ICLR.

---

### Official Review · AnonReviewer1 · 2018-11-02
**neat idea but not convincing enough**

**Rating:** 2
**Confidence:** 5

**Review:**

The authors propose a modified ReLU, the GaLU, where the nonlinearity gating role is decoupled from the linear weights. Similar ideas have been previously proposed. For example Tsai et al: http://papers.nips.cc/paper/6516-tensor-switching-networks: "The TS network decouples a hidden unit’s decision to activate (as encoded by the activation weights) from the analysis performed on the input when the unit is active (as encoded by the analysis weights)" and Veness et al: Online learning with gated linear networks, https://arxiv.org/abs/1712.01897.

In short, the paper proposes a tweak to the nonlinearity in neural nets. Since many tweaks have been previously investigated, for such a paper to be worthy of publication, in 2018, the experimental results need to be extremely impressive. The results in this paper, on MNIST and fashion-MNIST are nowhere near sufficient.

---

### Official Review · AnonReviewer3 · 2018-11-05
**Review of "Decoupling Gating from Linearity"**

**Rating:** 3
**Confidence:** 4

**Review:**

The paper introduces a GaLU activation function, which is the product of a random gate function and a learnable linear function. The authors argue that empirically, neural networks with the GaLU activation is as effective as that with the ReLU activation, but theoretically, the GaLU activation is easier to understand because of the separation of the non-linearity and the learnable parameters. The the paper analyzes neural networks with one GaLU layer. Essentially, the network is a random transformation followed by a linear projection. This property enables analysis that are well known for the linear models.

Although the definition of GaLU is new, the idea of combining a non-linear projection with a linear transformation is an old one. [1] shows that many kernel SVM models can be written in this form. [2] [3] show that neural networks with various of activation functions can be relaxed to this form. However, these methods have never achieved performance that is as good as the state-of-the-art CNN models in challenging datasets (ImageNet or even CIFAR-10).

In section 3, the accuracies on MNIST (98%) and MNIST-fashion (88%) are quite low. They are not even as good as a classical kernel SVM, though the non-linear projection version of the kernel SVM has been well-studied in theory.

In section 4, the analyses are mostly standard for linear models and convex optimization. To the best of our knowledge, it doesn't introduce new insight on the understanding of non-convex optimization.

Overall, I think the paper and its theoretical analysis is built on an unsolid claim that the GaLU activation is a good replacement for traditional non-linear activation functions. The empirical study doesn't seem to support this claim. I cannot recommend accepting the paper.

[1] Random Features for Large-Scale Kernel Machines
[2] Learning Kernel-Based Halfspaces with the Zero-One Loss
[3] Convexified Convolutional Neural Networks

---

### Meta-Review · Area_Chair1 · 2018-12-18

**Confidence:** 4
**Recommendation:** Reject

**Metareview:**

The reviewers reached a consensus that the paper is not ready for publication in ICLR. (see more details in the reviews below. )